# DISTA: Denoising Spiking Transformer with intrinsic plasticity and spatiotemporal attention

## Abstract

Among the array of neural network architectures, the Vision Transformer (ViT) stands out as a prominent choice, acclaimed for its exceptional expressiveness and consistent high performance in various vision applications. Recently, the emerging Spiking ViT approach has endeavored to harness spiking neurons, paving the way for a more brain-inspired transformer architecture that thrives in ultra-low power operations on dedicated neuromorphic hardware. Nevertheless, this approach remains confined to spatial self-attention and doesn't fully unlock the potential of spiking neural networks. We introduce DISTA, a Denoising Spiking Transformer with Intrinsic Plasticity and SpatioTemporal Attention, designed to maximize the spatiotemporal computational prowess of spiking neurons, particularly for vision applications. DISTA explores two types of spatiotemporal attentions: intrinsic neuron-level attention and network-level attention with explicit memory. Additionally, DISTA incorporates an efficient nonlinear denoising mechanism to quell the noise inherent in computed spatiotemporal attention maps, thereby resulting in further performance gains. Our DISTA transformer undergoes joint training involving synaptic plasticity (i.e., weight tuning) and intrinsic plasticity (i.e., membrane time constant tuning) and delivers state-of-the-art performances across several static image and dynamic neuromorphic datasets. With only 6 time steps, DISTA achieves remarkable top-1 accuracy on CIFAR10 (96.26%) and CIFAR100 (79.15%), as well as 79.1% on CIFAR10-DVS using 10 time steps.

## 1 Introduction

Originally designed for natural language processing applications (Vaswani et al., 2017), transformers have now gained popularity in various computer vision tasks, including image classification (Dosovitskiy et al., 2021), object detection (Carion et al., 2020), and semantic segmentation (Xie et al., 2021). Self-attention, a key component of transformers, selectively focuses on relevant information, enabling the capture of long-range interdependent features.

Spiking neural networks (SNNs), as the third generation of neural networks, harness efficient temporal coding and offer a higher level of biological plausibility compared to non-spiking artificial neural networks (ANNs). Moreover, SNNs can achieve significantly enhanced energy efficiency when deployed on specialized ultra-low-power neuromorphic hardware (Furber et al., 2014; Davies et al., 2018; Lee et al., 2022).

Hence, the logical progression is towards the development of SNN-based transformer architectures (Zhou et al., 2023b; Zhang et al., 2022; Zhu et al., 2023; Zhou et al., 2023a). Specifically, Zhou et al. (2023b;a) introduced a spike-based self-attention mechanism. This self-attention mechanism captures correlations between spatial patches occurring at the same time point, showing promising performance results. However, the spiking transformer of Zhou et al. (2023b;a) has several limitations. Firstly, the self-attention it employs is exclusively *spatial* in nature, involving only patches within the same time step. This constraint limits the network's expressive power. Conversely, the hallmark of SNNs lies in their ability to perform spatiotemporal computations, which can significantly enhance the performance of spiking transformers.

In pursuit of this goal, we propose a more comprehensive *spatiotemporal self-attention*, either intrinsically embedded in the basic operation of spiking neurons or through deliberate network architecture design. We explore two types of spatiotemporal attention mechanisms: intrinsic neural-level attention and explicitly designed network-level attention. Furthermore, in Zhou et al. (2023b;a), attention maps are formed by multiplying the binary spiking sequence queries ($Q$) with the keys ($K$), which is not noise-free. Passing such maps directly through without noise suppression may inadvertently downgrade the performance.

To address the aforementioned issues, we propose an architecture called Denoising spiking transformer with Intrinsic plasticity and SpatioTemporal Attention (DISTA) with three crucial elements, particularly for vision applications:

• **Neuron-Level Spatiotemporal Attention**: We demonstrate that by optimizing the intrinsic memory of individual neurons, we can elicit distinct responses to input spikes originating from different input neurons and occurring at various time intervals. This optimization results in neuron-level spatiotemporal attention, achieved through the tuning of each neuron's membrane time constant via intrinsic plasticity.

• **Network-Level Spatiotemporal Attention**: Furthermore, we introduce a network-level spatiotemporal attention mechanism, which leverages explicit memory and transcends the spatial-only attention of Zhou et al. (2023b;a). This network-level attention facilitates long-range attention computations that encompass firing activities occurring both in time and space.

• **Spatiotemporal Attention Denoising**: Lastly, we incorporate a non-linear denoising layer designed to mitigate noisy signals within the computed spatiotemporal attention map. This addition enhances the expressiveness of the spiking transformer, introducing both nonlinearity and noise suppression. We have conducted extensive experiments to validate the effectiveness of our proposed DISTA spiking transformer. The results, obtained across various static image and dynamic neuromorphic datasets, consistently demonstrate the superior performance of the DISTA architecture in comparison to prior spiking transformer approaches.

## 2 RELATED WORK

**Leaky Integrate-and-Fire (LIF) Neuron Models.** In contrast to conventional ANNs that operate on continuous-valued inputs and activations, SNNs utilize discrete binary spike sequences, generated by dynamic neuron models, for information computation and transmission. The LIF neuron model (Gerstner & Kistler, 2002), describing the dynamics of spiking neurons, has been adopted widely in recent SNN architectures. To emulate a spiking neuron, much research uses the fixed-step zero-order forward Euler method to discretize continuous membrane voltage updates over a set of discrete timesteps.

The resulting discrete-time LIF model is described by 2 variables: the neuronal membrane potential $V \in \mathbb{R}$, and the output spike sequence $s \in \{0, 1\}$. The intrinsic hyperparameters of a neuron include a decaying time constant of membrane potential $\tau_m \in \mathbb{R}$, and a firing threshold $\theta \in \mathbb{R}$. For neuron $i$ in layer $l$ of the network, at timestamp $t$, the membrane voltage $V_i^{(l)}[t]$ at timestamp $t$ and the output spike $s_i^{(l)}[t]$ are defined as:

$$V_i^{(l)}[t] = (1 - \frac{1}{\tau_m})V_i^{(l)}[t-1](1 - s_i^{(l)}[t-1]) + \sum_{j=1}^{D^{(l-1)}} w_{ij}^{(l)} s_j^{(l-1)}[t], \tag{1}$$

$$s_i^{(l)}[t] = H(V_i^l[t] - \theta), \tag{2}$$

where $H(\cdot)$ is the Heaviside step function.

SNNs can be trained by converting them into an approximately equivalent ANN or via direct training using backpropagation through time (BPTT) (Shrestha & Orchard, 2018; Wu et al., 2018; Jin et al., 2018; Zhang & Li, 2020; Kim et al., 2020; Yang et al., 2021).

**Spiking Transformers.** Several studies have undertaken investigations into transformer-based spiking neural networks for tasks such as image classification (Zhou et al., 2023b), object tracking

(Zhang et al., 2022), and the utilization of large language models (LLMs) (Zhu et al., 2023). Specifically, Zhang et al. (2022) has introduced a non-spiking ANN based transformer designed to process spiking data generated by Dynamic Vision Sensors (DVS) cameras. On the other hand, Zhou et al. (2023b) and Zhou et al. (2023a) have proposed a spiking vision transformer while incorporating only spatial self-attention mechanisms. Zhu et al. (2023) developed an SNN-ANN fusion language model, integrating a transformer-based spiking encoder with an ANN-based GPT-2 decoder to enhance the operational efficiency of LLMs.

However, while the aforementioned spiking transformer models have made significant contributions to the field, they have not fully explored the potential of spatiotemporal self-attention mechanisms or addressed the issue of noise suppression in attention maps, both of which are central to the research presented in this work.

## 3 METHOD

The conventional non-spiking Vision Transformer (ViT) (Dosovitskiy et al., 2021) consists of patch-splitting modules, encoder blocks, and linear classification heads. Each encoder block includes a self-attention layer and a multi-layer perceptron (MLP) layer. Self-attention empowers ViT to capture global dependencies among image patches, thereby enhancing feature representation (Katharopoulos et al., 2020).

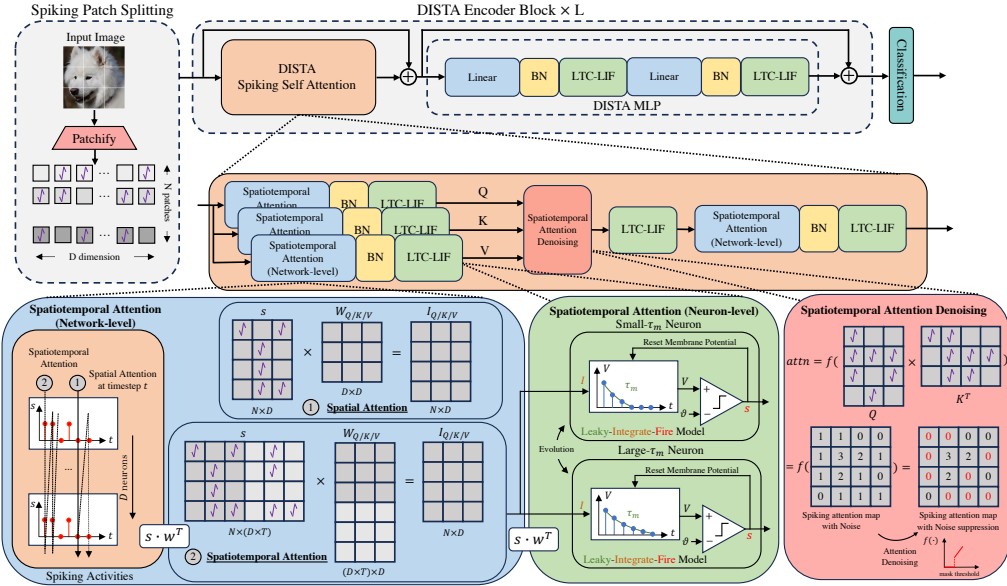

Figure 1: The overview of denoising spiking transformer with intrinsic plasticity and spatiotemporal attention: DISTA.

Prior spiking transformer models have adapted the architecture of ViT by incorporating a spiking patch-splitting module and processing a feature map with dimensions $N \times D$ over $T$ time steps (Zhou et al., 2023a;b) using spiking neurons. Among other adaptations, these models utilize spike-based multiplication to compute spatial-only attention maps in the form of $Q[t]K^T[t]$ for each time step, replacing the non-spiking counterparts $QK^T$ in the original ViT models, where $Q$ and $K$ represent "query" and "key," respectively.

The proposed DISTA spiking transformers, as illustrated in Figure 1, follow the fundamental network architecture of previous spiking transformers (Zhou et al., 2023a;b). The core of the DISTA architecture consists of $L$ DISTA encoder blocks, each comprising several layers. Notably, within each encoder block, we introduce several key innovations. Firstly, we employ spatiotemporal attention at both the network and neuron levels, the latter facilitated by intrinsic plasticity-based tuning of neuron membrane time constants. Secondly, we enhance performance by suppressing noise in each attention map through the application of a nonlinear denoising function $f$, resulting in a de-

noised map $A = f(Q[t]K^T[t])$. The product of $A$ and the spike-based value $V$ is then passed to the subsequent layer in the encoder block.

## 3.1 NEURON-LEVEL SPATIOTEMPORAL ATTENTION

The initial category of spatiotemporal attention mechanisms we investigate involves neuron-level attention, which is integral to the functioning of spiking neurons, as illustrated in Figure 2. We fine-tune these neuron-level spatiotemporal attention mechanisms for a specific visual task by adjusting the membrane time constant ($\tau_m$) of each spiking neuron.

In Figure 2, we observe that neuron 3 receives inputs from neurons 1 and 2, with these inputs originating from different spatial positions and occurring at distinct time points. Due to the inherent memory in its membrane potential state, neuron 3 serves as an intrinsic computational unit attending to the firing activities of neurons 1 and 2. It's important to note that the dynamic characteristics of the membrane potential play a crucial role in computing neuron-level attention. When the membrane time constant ($\tau_m$) is set to a higher value, the membrane potential of neuron 3 decays more slowly, effectively creating a longer time window for processing the attention of the received spike inputs. This extended time period allows neuron 3 to potentially fire one or multiple times in response to the input spikes from neurons 1 and 2. On the other hand, a smaller $\tau_m$ for neuron 3 results in weaker retention of past spike inputs. The faster decay of the membrane potential makes it less likely for neuron 3 to fire, thereby reducing its attentiveness to the received spatiotemporal inputs.

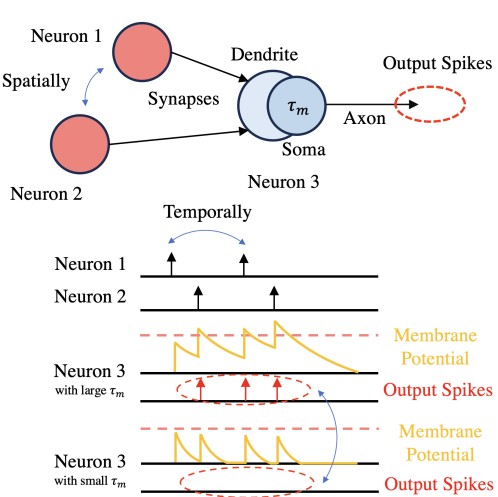

Figure 2: Neuron-level Spatiotemporal Attention.

To optimize neuron-level spatiotemporal attention for a given learning task, we employ intrinsic plasticity to fine-tune the critical intrinsic parameter, $\tau_m$, for each spiking neuron. In addition to the conventional approach of adjusting weight parameters (synaptic plasticity), we make each spiking neuron's membrane time constant a learnable parameter, referred to as Learnable Membrane Time Constants (LTC). We integrate the optimization of these membrane time constants into SNN-based backpropagation, specifically Backpropagation Through Time (BPTT), in order to tailor neuron-level spatiotemporal attention. The backpropagation of $\tau_m$ is achieved through the following update rule: $\tau_{m,(i)}^{(l)} = \tau_{m,(i)}^{(l)} - \alpha \frac{\partial L}{\partial \tau_{m,(i)}^{(l)}}$, where $\tau_{m,(i)}^{(l)}$ represents the membrane time constant of neuron $i$ within layer $l$, $\alpha$ denotes the learning rate, and $L$ signifies the loss function.

## 3.2 NETWORK-LEVEL SPATIOTEMPORAL ATTENTION

As depicted in Figure 1, we apply the same network-level spatiotemporal mechanisms to two different layers within each encoder block. We will elaborate on how this scheme is utilized to calculate the spike inputs for the three LTC-LIF (LIF neurons with learnable time constants) spiking neuron arrays, whose output activations define the query ($Q$), key ($K$), and value ($V$).

In Zhou et al. (2023a;b), at each time point $t$, the spike inputs to the current $l$-th encoder block, which are the outputs $s^{(l-1)}[t]$ of the previous $(l-1)$-th encoder block, undergo multiplication with corresponding weights to yield the inputs $I_{Q/K/V}[t]$ to the query/key/value LTC-LIF neuron arrays: $I_{Q/K/V}[t] = s^{(l-1)}[t] \times W_{Q/K/V}$, where $I_{Q/K/V}[t]$ and $s^{(l-1)}[t] \in R^{N \times D}$, and $W_{Q/K/V} \in R^{D \times D}$. Each entry in $I_{Q/K/V}[t]$ corresponds to the input to a specific neuron in the query/key/value arrays, and we can see that it only retains the information of spike outputs of $D$ neurons from the $(l-1)$-th encoder block at time $t$ (i.e., a row of $s^{(l-1)}[t]$). Hence, we refer to the resulting attention as "spatial only", as illustrated in ❶ of Figure 1.

Notably, in DISTA, we transform the attention from "spatial-only" to "spatiotemporal," as illustrated in ❷ of Figure 1. To achieve this, we leverage not only the spiking activities of these $D$ neurons at time $t$ but also those occurring before $t$ while adhering to the causality of spiking activities (Hebb, 1949). This approach forms the input to each subsequent query/key/value LTC-LIF neuron. Specifically, the input to the query/key/value neuron at location $(i, k)$ is expressed as follows:

$$I_{ik}^{(l)}[t] = \sum_{m=0}^{t} \sum_{k=0}^{D} w_{jkm}^{(l)} s_{ik}^{(l-1)}[m]. \tag{3}$$

With the above synaptic input, each query/key/value neuron is emulated by the following discretized leaky integrate-and-fire dynamics:

$$V_{ik}^{(l)}[t] = (1 - \frac{1}{\tau_m})V_{ik}^{(l)}[t = 1](1 - s_{ik}^{(l)}[t - 1]) + I_{ik}^{(l)}[t], \tag{4}$$

$$s_{ik}^{(l)}[t] = H(V_{ik}^{l}[t] - \theta). \tag{5}$$

### 3.2.1 TEMPORAL ATTENTION WINDOW (TAW)

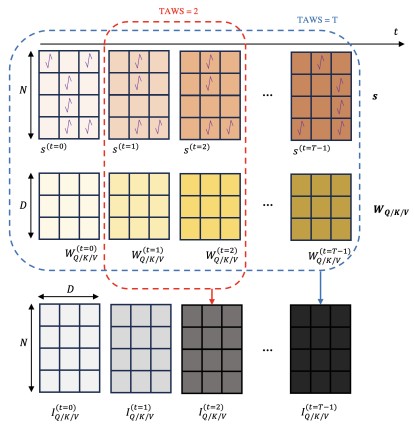

Figure 3: Network-level spatiotemporal attention within a given temporal attention window.

When constructing the spatiotemporal inputs for the query/key/value neurons, as defined in Eq 3, we have the option to control the extent of temporal attention by imposing a temporal attention window (TAW). This serves as an optimization, allowing us to balance computational resources and performance. When using a TAW size of 2, we reference both the present and previous time step to create the inputs for the query/key/value neuron arrays, i.e., $I_{Q/K/V}^{(t=2)} = s^{(t=1)} \times W_{Q,K,V}^{t=1} + s^{(t=2)} \times W_{Q,K,V}^{(t=2)}$. A TAW size of $T$ represents the complete Temporal Attention Window Size (TAWS). In this scenario, all the spike inputs received at the encoder block level up to the current time step are utilized to construct the inputs for the query/key/value neurons: $I_{Q/K/V}^{(t=T-1)} = \sum_{t=0}^{T-1} s^{(t)} \times W_{Q,K,V}^{(t)}$. In general, we have observed that increasing the Temporal Attention Window Size (TAWS) initially enhances performance, but this improvement tends to plateau after a certain point as demonstrated in Appendix B.

### 3.3 SPATIOTEMPORAL ATTENTION DENOISING

In the attention layers of existing spiking transformers (Zhou et al., 2023b;a), a timestep-wise spiking attention map is generated by multiplying the outputs of the query neuron array ($Q$) with those of the key neuron array ($K$). Each entry in this map corresponds to a pairing of query and key neurons, where a one-to-one spatial correspondence is maintained. It's important to note that a nonzero value in the attention map signifies the simultaneous activation of one or multiple query-key neuron pairs. However, the computed spike-based attention maps are not necessarily devoid of noise, and the existing spiking transformers lack efficient noise suppression mechanisms.

To address this concern, we introduce a nonlinear denoising mechanism. Nonlinearity is paramount in enhancing the capabilities of deep neural networks by enabling them to capture intricate relationships between variables (LeCun et al., 2015). In conventional ANN-ViTs, a non-linear softmax function is widely employed to filter the attention map, a practice not adopted in existing spiking transformers due to concerns about computational complexity. Instead, the primary source of nonlinearity in spiking transformers stems from the utilization of leaky integrate-and-fire (LIF) neurons,

which, while effective, possess limited expressive power. To overcome this limitation, we introduce an A̲ttention D̲e̲N̲osing(ADN) Mechanism, seamlessly integrated within the DISTA spiking self-attention layers, as depicted in Figure 1.

Mathematically, to denoise a $N \times N$ spatiotemporal attention map, we assign a value of zero to any entry in the attention map that falls below a specified threshold $u$:

$$a_{t,i,j}^{(l,masked)} = \begin{cases} 0 & \text{if } a_{t,i,j}^{(l)} < u^{(l)} \\ a_{t,i,j}^{(l)} & \text{if } a_{t,i,j}^{(l)} \geq u^{(l)} \end{cases} \qquad (6)$$

In Equation 6, $a_{t,i,j}^{(l)}$ denotes the correlation between patches $i$ and $j$ at timestep $t$ within block $l$'s spiking attention layer.

It's worth noting that ADN can be efficiently implemented in hardware using comparators within the attention layer. Additionally, it has a space and time complexity of $O(N^2)$, which is notably more efficient when contrasted with the $O(N^2D)$ complexity associated with matrix multiplication operations within the encoder blocks.

## 4 EXPERIMENTS

We assess the performance of our DISTA spiking transformer by comparing it with existing SNN networks trained using various methods and the recent spiking transformer employing spatial-only attention (Zhou et al., 2023b) based on several popular image and dynamic neuromorphic dataset, commonly adopted for evaluating spiking neural networks.

### 4.1 RESULTS ON CIFAR10/100 IMAGE DATASETS

**CIFAR10 and CIFAR100** The CIFAR datasets (Krizhevsky, 2009) contain 50,000 training images and 10,000 testing images, 10 and 100 categories, respectively, and are commonly adopted for testing directly trained SNNs. The pixel resolution of each image is 32x32. A spiking patch splitting module split each image into 64 $4 \times 4$ patches. We employ a batch size of 256 or 512 and adopt the AdamW optimizer to train our DISTA spiking transformer over 1,000 training epochs, and compare it with several baselines. A standard data augmentation method, such as random augmentation, mixup, or cutmix is also used in training for a fair comparison. The learning rate is initialized to 0.003 with a scaling factor of 0.125 for cosine decaying as in Zhou et al. (2023b). We initialize $\tau_m$ of all spiking neurons to be 2, and the denoising threshold to be 3 for all layers.

| Model/Traing Method | Architecture | Timesteps | Accuracy |
|---|---|---|---|
| TSSL-BP(Zhang & Li, 2020) | CIFARNet | 5 | 91.41% |
| STBP-tdBN(Zheng et al., 2021) | CIFARNet | 5 | 92.92% |
| NA(Yang et al., 2021) | AlexNet | 5 | 91.76% |
| Hybrid training(Rathi et al., 2020) | VGG-11 | 125 | 92.22% |
| TET(Deng et al., 2022) | ResNet-19 | 4 | 94.44% |
| DT-SNN(Li et al., 2023) | ResNet-19 | 4 | 93.87% |
| Diet-SNN(Rathi & Roy, 2020) | ResNet-20 | 5 | 92.54% |
| Spikformer(Zhou et al., 2023b) | Spikformer-4-384 | 6 | 95.34% |
| Spikformer(Zhou et al., 2023b) | Spikformer-4-384 | 4 | 95.19% |
| DISTA | Spikformer-4-384 | 6 | **96.26%** |
| | Spikformer-4-384 | 4 | **96.10%** |

Table 1: Comparison of the DISTA spiking transformer with other SNNs on CIFAR10

We compare the performance of our DISTA model on CIFAR10 with the baseline spiking transformer model employing spatial-only attention (Zhou et al., 2023b), and other SNNs based on CIFARNet, AlexNet, VGG-11 and ResNet-19 as shown in Table 1. Based on the transformer's architecture with 384-dimensional embedding and four encoders (Spikformer-4-384) in Zhou et al. (2023b), DISTA gains a significant improvement on top-1 accuracy on CIFAR10, improving the baseline spiking transformer's accuracy from 95.34% to 96.26% and from 95.19% to 96.10% when

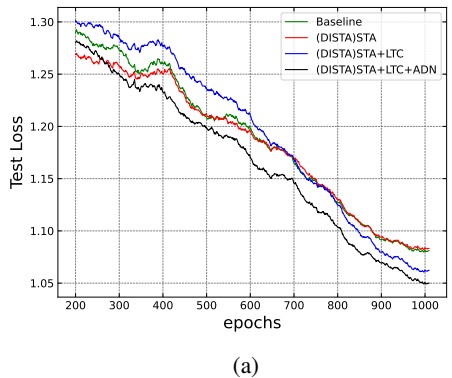 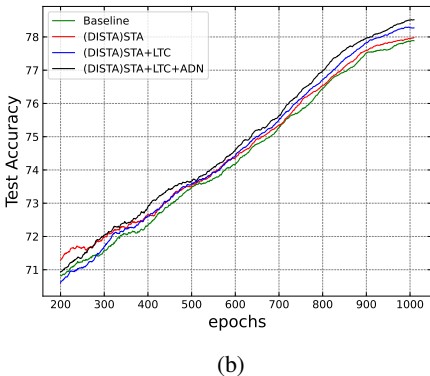

(a)                  (b)

Figure 4: Training convergence of the DISTA-4-384-6 spiking Transformer vs. the baseline (Zhou et al., 2023b) on CIFAR100: (a) test loss, and (b) test accuracy. The solid curves are 25-epoch moving averages.

the number of timesteps is 6 and 4, respectively; Executing over 4 time steps, DISTA obtains a significant accuracy improvement of 1.66% or more over ResNet-19, and improves by 3.56% the accuracy of the ResNet-20 model with 5 timesteps.

We evaluate our DISTA model on CIFAR100 in Table 2, where the * subscript highlights our reproduced results. Based on the same spikformer-4-384 architecture, DISTA gains noticeable accuracy improvements over these SNNs and the baseline spiking transformer when executed over 4 or 6 time steps.

| Model/Training Method | Architecture | Timesteps | Accuracy |
|---|---|---|---|
| Hybrid training(Rathi et al., 2020) | VGG-11 | 125 | 67.87% |
| TET(Deng et al., 2022) | ResNet-19 | 4 | 74.47% |
| STBP-tdBN(Zheng et al., 2021) | ResNet-19 | 4 | 74.47% |
| TET(Deng et al., 2022) | ResNet-19 | 4 | 74.47% |
| DT-SNN(Li et al., 2023) | ResNet-19 | 4 | 73.48% |
| Diet-SNN(Rathi & Roy, 2020) | ResNet-20 | 5 | 64.07% |
| Spikformer[*](Zhou et al., 2023b) | Spikformer-4-384 | 6 | 78.36% |
| Spikformer[*](Zhou et al., 2023b) | Spikformer-4-384 | 4 | 78.04% |
| DISTA | Spikformer-4-384 | 6 | **79.15%** |
| | Spikformer-4-384 | 4 | **78.77%** |

Table 2: Comparison of the DISTA spiking transformer with other SNNs on CIFAR100.

Figure 4 illustrates the training dynamics of DISTA, where "STA" represents network-level spatiotemporal attention, "LTC" indicates learnable membrane time constants, and "ADN" denotes attention denoising. Notably, DISTA (depicted by the black line) exhibits superior performance compared to the baseline Zhou et al. (2023b) in the middle and later stages of the training process. Additionally, we observe that the inclusion of LTC enhances the performance of STA-only, highlighting the benefits of spatiotemporal attentions at both the neuron and network levels. This performance is further enhanced by the inclusion of ADN.

## 4.2 RESULTS ON DYNAMIC NEUROMORPHIC DATASET

**CIFAR10-DVS** CIFAR10-DVS (Li et al., 2017) is a neuromorphic dataset containing dynamic spike streams captured by a dynamic vision sensor (DVS) camera viewing moving images from the CIFAR10 datasets. It contains 9,000 training samples and 1,000 test samples. For this dataset, the optimizer used was AdamW, the batch size was set to 256 and the learning rate was set at 0.01 with a cosine decay. The neuromorphic data augmentation technique from Li et al. (2022) was applied. The training was performed over 300 epochs. The classification performances of DISTA and the

spiking transformer baseline, as well as other state-of-the-art SNN models are shown in Table 3. Again, our DISTA transformer outperforms all existing SNN models under various settings. In particular, DISTA outperforms the baseline spiking transformer by 3.7% and 6.1% when running over 4 and 8 time steps, respectively.

| Model/Training Method | Architecture | Timesteps | Accuracy |
|---|---|---|---|
| BNTT(Kim & Panda, 2020) | six-layer CNN | 20 | 63.2% |
| SALT(Kim & Panda, 2021) | VGG-11 | 20 | 67.1% |
| PLIF(Fang et al., 2021) | VGG-11 | 20 | 74.8% |
| Rollout(Kugele et al., 2020) | VGG-16 | 48 | 66.5% |
| NDA(Li et al., 2022) | ResNet-19 | 10 | 78.0% |
| DT-SNN(Li et al., 2023) | ResNet-19 | 10 | 74.8% |
| Spikformer[*](Zhou et al., 2023b) | Spikformer-4-384 | 4 | 73.2% |
| Spikformer[*](Zhou et al., 2023b) | Spikformer-4-384 | 8 | 73.9% |
| Spikformer(Zhou et al., 2023b) | Spikformer-4-384 | 10 | 78.9% |
| DISTA | Spikformer-4-384 | 4 | **76.9%** |
| | Spikformer-4-384 | 8 | **79.0%** |
| | Spikformer-4-384 | 10 | **79.1%** |

Table 3: Comparison of the DISTA spiking transformer with other SNNs on CIFAR10.

## 4.3 ABLATION STUDY

**Proposed key elements in DISTA.** We analyze the effect of each proposed key element in the DISTA transformer on CIFAR100 using 4 or 6 timesteps in Table 4: (1) STA (Network-level SpatioTemporal Attention Only), (2) LTC + STA (Learnable Membrane Time Constant + Network-level SpatioTemporal Attention), and (3) STA + LTC + ADN (Network-level SpatioTemporal Attention + Learnable Membrane Time Constant + Attention DeNoising). Overall, these proposed techniques can lead to further performance improvements of the DISTA spiking transformer.

| Model | Architecture | Timesteps | Accuracy |
|---|---|---|---|
| DISTA (w/ ST-attention) | Spikformer-4-384 | 6 | 78.58% |
| DISTA (w/ ST-attention+LTC) | Spikformer-4-384 | 6 | 78.87% |
| DISTA (w/ ST-attention+LTC+ADN) | Spikformer-4-384 | 6 | 79.15% |
| DISTA (w/ ST-attention) | Spikformer-4-384 | 4 | 78.40% |
| DISTA (w/ ST-attention+LTC) | Spikformer-4-384 | 4 | 78.11% |
| DISTA (w/ ST-attention+LTC+ADN) | Spikformer-4-384 | 4 | 78.77% |

Table 4: Ablation study on different elements of DISTA on CIFAR100.

| Dataset | Model | Timesteps | Accuracy |
|---|---|---|---|
| CIFAR100 | DISTA-Spikformer-4-384 | 1 | 74.43% |
| | | 2 | 76.65% |
| | | 4 | 78.77% |
| | | 6 | 79.15% |

Table 5: Ablation study on number of simulation timesteps on CIFAR100 dataset.

**Number of Timesteps.** We investigate the influence of the number of simulation time steps as detailed in Table 5. It can be seen that the performance of the DISTA transformer steadily increases with the number of timesteps at the cost of increasing computational overhead. Running over a single time eliminates temporal processing in the model. Under this case, DISTA achieves an accuracy of 74.43%, surpassing the 74.36% performance of the baseline spiking transformer (Zhou et al., 2023b). This improvement can primarily be attributed to the incorporation of learnable membrane time constants and the proposed attention-denoising mechanism. When the number of timesteps increased to 2, DISTA demonstrates an accuracy of 76.65%, outperforming the baseline spiking transformer's 76.28% accuracy. The presented results underline the consistent ability of DISTA to improve the transformer's performance, especially in scenarios of low time step count.

**Attention Denoising.** We delve into the impact of denoising setting on performance in Table 6. The effect of the denoising threshold ($u$) per Eq 6 is evaluated on CIFAR100. We observe that the transformer's performance continues to improve until $u$ reaches 4. On CIFAR10, we analyze the influence of the number of encoder blocks ($b$) with denoising activated. In general, applying denoising to a larger number of encoder leads to a better performance.

| Datasets | Architecture | Timesteps | Denoising Setting | Accuracy |
|----------|--------------|-----------|-------------------|----------|
| CIFAR100 | Spikformer-4-384 | 6 | w/o ADN | 77.21% |
|          |              |   | ADN-truncation($u = 2$) | 77.26% |
|          |              |   | ADN-truncation($u = 3$) | 77.32% |
|          |              |   | ADN-truncation($u = 4$) | 77.14% |
| CIFAR10  | Spikformer-4-384 | 4 | w/o ADN | 95.90% |
|          |              |   | ADN-truncation($b = 1$) | 96.16% |
|          |              |   | ADN-truncation($b = 2$) | 96.26% |
|          |              |   | ADN-truncation($b = 3$) | 96.25% |

Table 6: Ablation study on different denoising settings on CIFAR10 and CIFAR100.

## 5 CONCLUSION

In this work, we introduce DISTA, a denoising spiking transformer that incorporates intrinsic plasticity and spatiotemporal attention. We place a particular emphasis on exploring spatiotemporal attention mechanisms within spiking transformers, both at the neuron level and at the network level. These mechanisms enable spiking transformers to seamlessly fuse spatiotemporal attention information with aid of backpropagation-based synaptic and intrinsic plasticity. Furthermore, we have introduced a denoising technique for attenuating the noise in the computed attention maps, which improves performance while introducing very low additional computational overhead. Our DISTA spiking transformer outperforms the current state-of-the-art SNN models on several image and neuromorphic datasets. We believe that our investigations provide a promising foundation for future research in the domain of computationally efficient high-performance SNN-based transformer models.

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

# A    MULTIPLE HEAD ATTENTION

To implement multiple head attention, we divide $Q$, $K$, and $V$ into H parts and process them simultaneously using H instances of DISTA Spiking Self Attention (DSSA), referred to as H-head DSSA. The multiple-head DSSA(MDSSA) is shown as follows:

$$Q = (q_0, q_1, ..., q_{H-1}), K = (k_0, k_1, ..., k_{H-1}), V = (v_0, v_1, ..., v_{H-1}), q, k, v \in \mathbb{R}^{T \times N \times d} \quad (7)$$

where, $Q, K, V \in \mathbb{R}^{T \times N \times D}, D = H \times d$.

$$MDSSA^{'}(Q, K, V) = [DSSA_0(q_0, k_0, v_0), ..., DSSA_0(q_{H-1}, k_{H-1}, v_{H-1})] \quad (8)$$

$$MDSSA(Q, K, V) = LIF_{LTC}(BatchNorm(Linear(MDSSA^{'}(Q, K, V)))) \quad (9)$$

In CIFAR100 and CIFAR100, we apply 12-head spiking self-attention in all experiments; in CIFAR10-DVS, we apply 16-head self-attention.

# B    THE EFFECT OF TEMPORAL ATTENTION WINDOW SIZE

We investigate the effect of temporal attention window size on performance in Figure 5. The model is DISTA-Spikformer-4-384 with 6 timesteps on CIFAR10, we find that under two different fixed membrane time constants ($\tau_m$), a larger coverage ratio causes a better performance, but saturation when it exceeds certain ratio.

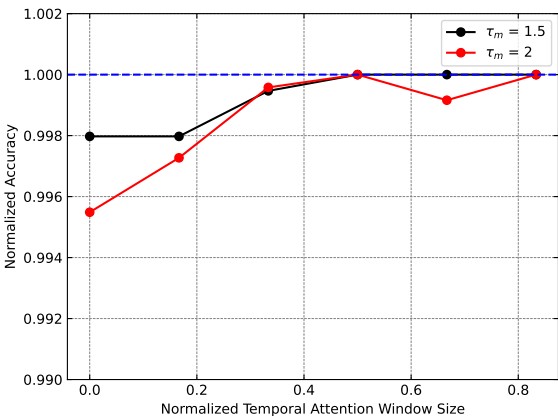

Figure 5: The effect of temporal attention window size on the accuracy of the proposed DISTA spiking transformer evaluated on CIFAR10.

