# OpenReview forum: "DISTA: DENOISING SPIKING TRANSFORMER WITH INTRINSIC PLASTICITY AND SPATIOTEMPORAL ATTENTION"
_ICLR.cc/2024/Conference — Submitted to ICLR 2024_

### Official Review · Reviewer_kKsP · 2023-10-28

**Soundness:** 2 fair
**Presentation:** 2 fair
**Contribution:** 2 fair
**Rating:** 3
**Confidence:** 5

**Summary:**

this work introduces a Denoising Spiking Transformer with Intrinsic Plasticity and SpatioTemporal Attention. The work did experiments on Cifar.

**Strengths:**

1. the work provides two types of spatiotemporal attentions: intrinsic neuron-level attention and network-level attention.
2.  the work also provides an efficient nonlinear denoising mechanism.

**Weaknesses:**

1. The main weakness of the paper is the novelty, the  SpatioTemporal Attention has been proposed in prior work[1].
2. The imagenet dataset is widely used in the SNN field, but the work misses it.
3. Other results in the SPIKFORMER should be compared in the work.
4. Some recent work is missing, e.g., [2,3]
[1]Spatial-Temporal Self-Attention for Asynchronous Spiking Neural Networks
[2]Surrogate Module Learning: Reduce the Gradient Error Accumulation in Training Spiking Neural Networks
[3]Spikingformer: Spike-driven Residual Learning for Transformer-based Spiking Neural Network

**Questions:**

see weakness.

---

> ### Author Response · Authors · 2023-11-23
> **Thank you for your insightful review!**
>
> Q1: The main weakness of the paper is the novelty, the SpatioTemporal Attention has been proposed in prior work[1].
>
> A1: We apologize for our lack of knowledge of the reference paper the reviewer provided during the development of our work.  We will reference it in the paper. Nevertheless,  we would like to highlight the key contributions of our paper.
>
> 1. Two-level spatiotemporal attention based on neuron-level intrinsic plasticity and network-level spatiotemporal attention: DISTA uniquely integrates the network-level spatiotemporal attention with the intrinsic plasticity mechanism, allowing neurons within the spiking neural network to adaptively adjust their firing thresholds. This feature significantly enhances the model's ability to process and learn from spatiotemporal data, setting it apart from previous models that do not include such a mechanism. As the reviewer has pointed out, the spirit of our network-level spatiotemporal attention is similar to that of the reference paper [1]. Nevertheless, we want to present the difference between our approach and [1]. In [1], spatiotemporal attention is introduced when computing the attention map using keys and queries. Our approach, however, considers spatiotemporal attention when computing keys, queries, and values. As such, these two approaches are complementary. The benefit of the spatiotemporal attention scheme of [1] can be leveraged by our work, which may further improve accuracy.
>
> 2. Denoising Approach: DISTA is distinctively designed with a denoising approach tailored for spiking neural networks. This aspect of the architecture aids in reducing the impact of spiking noise, thereby enhancing the robustness and performance of the model, especially in scenarios with varying data quality. In addition, we enable the attention map non-linearity by our proposed denoising method, in which we truncate the values that are smaller than the noise threshold to zero. Furthermore, denoising significantly reduces hardware overhead in the attention layers as shown in the general responses.
>
> Q2: The ImageNet dataset is widely used in the SNN field, but the work misses it.
>
> A2: We conducted an experiment on ImageNet-100 and presented the results in the general responses at the very beginning.
>
> Q3: Other results in the spikformer should be compared in the work.
>
> A3: We added a detailed hardware energy analysis  in the general responses at the very beginning.
>
> Q4: Some recent work is missing, e.g., [2,3]
>
> A4: Thank you for pointing these out. We will add a review of these works to our paper!
>
> [1]Spatial-Temporal Self-Attention for Asynchronous Spiking Neural Networks
> [2]Surrogate Module Learning: Reduce the Gradient Error Accumulation in Training Spiking Neural Networks
> [3]Spikingformer: Spike-driven Residual Learning for Transformer-based Spiking Neural Network

---

### Official Review · Reviewer_5bGj · 2023-10-31

**Soundness:** 2 fair
**Presentation:** 4 excellent
**Contribution:** 2 fair
**Rating:** 3
**Confidence:** 4

**Summary:**

The paper introduces DISTA, a novel neural network architecture that synergizes the strengths of Vision Transformer (ViT) with the potential of spiking neurons. DISTA stands out with its "Denoising Spiking Transformer" design that integrates Intrinsic Plasticity and Spatiotemporal Attention, optimizing its computational prowess for vision tasks. A distinct innovation lies in its noise-reducing mechanism for computed spatiotemporal attention maps, ensuring more refined performance. Demonstrating its efficacy, DISTA yields notable results on benchmark datasets such as CIFAR10 and CIFAR10-DVS.

**Strengths:**

1.	It changed the Linear layer into production of matrix, and the complexity is indeed O(ND^2) or O(TND^2).
2.	This work adopted PLIF to improve the performances.
3.	This work explores two types of spatiotemporal attentions.
4.	This paper is well written.

**Weaknesses:**

The biggest flaw is training for 1000 Epochs on CIFAR100 and CIFAR10.

The algorithm demonstrates improvements on performance on both CIFAR10 and CIFAR100 datasets; however, the improvement on CIFAR100 is not that significant, and it has not been assessed on larger datasets like ImageNet.

Lacking details on complexity, parameters etc.

**Questions:**

Why the model converge so slow (1000 Epochs)

How's the performance on ImageNet

Is there any improvements on PLIF or just grab it without any changes

Please add the analysis on complexity, parameters etc

---

> ### Author Response · Authors · 2023-11-23
> **Thank you for your insightful review!**
>
> Q1: Why does the model converge so slowly?
>
> A1: We would like to point out that slow convergence is a general property of transformers instead of a characteristic of our DISTA. For comparison, we use spikformer as a baseline to show the relationship between model performance and epoch number. It can be seen that our method has better results and faster convergence speed than spikformer at any epoch checkpoint. The table below shows model test accuracy(%) during training on CIFAR100.
> | Method\Epoch       | 200  | 300  | 400  | 500  | 600  | 700  | 800  | 900  | 1000 |
> |--------------|------|------|------|------|------|------|------|------|------|
> | Spikeformer  | 72.21| 72.96| 73.76| 74.29| 75.30| 76.17| 77.12| 78.13| 78.36|
> | DISTA        | 72.59| 73.02| 73.77| 74.66| 75.47| 76.57| 77.88| 78.85| 79.15|
>
> Q2: How's the performance on ImageNet?
>
> A2: We conducted an experiment on ImageNet-100 and presented the results in the general responses.
>
> Q3: Are there any improvements on PLIF or just grab it without any changes?
>
> A3: We adopt the Parameterized LIF model as a tool that provides a trainable membrane potential constant in SNNs to support our neuron-level spatiotemporal attention.  Our experiment proves that, together with our Network-level spatiotemporal attention mechanism and denoising function, a trainable membrane potential constant improves the performance of spiking transformers.
>
> Q4: Please add the analysis on complexity and parameters.
>
> A4: We provided a detailed hardware energy and space complexity analysis in the general responses at the very beginning.

---

### Official Review · Reviewer_DFpL · 2023-10-31

**Soundness:** 3 good
**Presentation:** 3 good
**Contribution:** 3 good
**Rating:** 6
**Confidence:** 4

**Summary:**

The paper introduces DISTA, a novel architecture for spiking neural networks (SNNs) aiming to improve their performance on vision tasks by utilizing spatiotemporal attention mechanisms.

**Strengths:**

+ Innovative Architecture: DISTA innovates by integrating spatiotemporal attention at both neuron and network levels, along with a denoising mechanism, which leads to state-of-the-art performance on several datasets.

+ Energy Efficiency: By leveraging the inherent efficiencies of SNNs and denoising mechanisms, DISTA promises to improve energy efficiency, which is crucial for neuromorphic computing.

+ Biologically Inspired: The approach takes inspiration from the biological brain, potentially opening up new avenues for understanding neural processing.

+ High Accuracy: DISTA achieves remarkable accuracy improvements on the CIFAR10 and CIFAR100 datasets, demonstrating the potential of SNNs in complex tasks.

+ Backpropagation Integration: The paper integrates backpropagation-based synaptic and intrinsic plasticity into SNNs, enhancing their learning capabilities.

**Weaknesses:**

- Novelty: I may have missed something, but most of the approach is taken from the spikeformer paper. This work mainly taken the temporal attention window which results in an increase in number of parameters and that may be the reason why the authors get accuracy improvement. I am not sure if the temporal attention has anything to do with it. If yes, it ll be good if the authors can elaborate. Further,there is no result on imagenet dataset. I feel the reason why authors didn't do it was the temporal attention ll explode the overall parameter requirement which becomes unmanageable in alarge dataset. On the other hand, the original spikeformer work has imagenet results. Kindly clarify the scalability of your approach or discuss its limitations.

- Specificity of Application: While DISTA shows significant improvements in vision tasks, its adaptability to other types of tasks is not demonstrated. Can the authors comment on this?

- Overfitting Risk: There's a potential risk of overfitting as the model complexity increases with multiple attention mechanisms, though this is not explicitly discussed in the paper. Can the authors provide their opinion on this?

- Can the authors do some off the shelf energy estimations maybe by using some spiking hardware simulator tools like [1] (its understandable if the authors are not able to do it fully, a qualitative discussion on the efficiency benefits- as in whether they expect sparsity advantage at compute level or memory storage reduction from a hardware point of view will be useful) ? I will be interested to see if their DISTA is actually energy efficient (specifically in terms of memory energy).

[1] https://github.com/RuokaiYin/SATA_Sim

**Questions:**

see weaknesses above. I am giving a rating of 6 mainly because of novelty concerns. If the authors can provide suitable justification, I ll change my rating.

---

> ### Author Response · Authors · 2023-11-23
> **Thank you for your insightful review!**
>
> Q1:  I may have missed something, but most of the approach is taken from the spikeformer paper. This work mainly taken the temporal attention window which results in an increase in the number of parameters and that may be the reason why the authors get accuracy improvement. I am not sure if the temporal attention has anything to do with it. If yes, it ll be good if the authors can elaborate. Further, there is no result on the ImageNet dataset. I feel the reason why authors didn't do it was the temporal attention ll explode the overall parameter requirement which becomes unmanageable in a large dataset. On the other hand, the original spikformer work has ImageNet results. Kindly clarify the scalability of your approach or discuss its limitations.
>
> A1: We conducted an experiment on ImageNet-100 and presented the results in the general responses.
>
> Q2: While DISTA shows significant improvements in vision tasks, its adaptability to other types of tasks is not demonstrated. Can the authors comment on this?
>
> A2: Although we only conducted experiments on visual tasks in the paper, our three core methods: neuron-level spatiotemporal attention, network-level spatiotemporal attention, and spatiotemporal attention denoising. They are in principle universal for all transformer architectures and are not limited to the visual realm. In principle, our method could be directly applied to spiking versions of transformers in other fields, such as Natural Language Processing (Bert, GPT), Reinforcement Learning (Decision Transformer), and Time Series Prediction (Temporal Fusion Transformers). We plan to explore some of these additional applications in our future work.
>
> Q3: There's a potential risk of overfitting as the model complexity increases with multiple attention mechanisms, though this is not explicitly discussed in the paper. Can the authors provide their opinion on this?
>
> A3: Although the extra parameters we added in DISTA brought the risk of overfitting, it did not actually happen. Theoretically, neuron-level spatiotemporal attention enhances the nonlinear processing capabilities of neurons, network-level spatiotemporal attention provides short-term memory of past spiking information, and spatiotemporal attention denoising provides noise reduction capabilities as an alternative to softmax. Experimentally, the decent test accuracy and learning curve also proved that overfitting did not occur.
>
> Q4: Can the authors do some off-the-shelf energy estimations maybe by using some spiking hardware simulator tools like [1] (it's understandable if the authors are not able to do it fully, a qualitative discussion on the efficiency benefits- as in whether they expect sparsity advantage at compute level or memory storage reduction from a hardware point of view will be useful)?
>
> A4: We provided a detailed hardware energy analysis in the general responses at the very beginning.

---

### Official Review · Reviewer_AvKg · 2023-10-31

**Soundness:** 2 fair
**Presentation:** 3 good
**Contribution:** 2 fair
**Rating:** 6
**Confidence:** 4

**Summary:**

The authors have tackled a rather interesting and important problem, which is to design/improve spiking neural networks-based vision transformers. More specifically, the authors have proposed a spiking transformer model that provides node-level and network-wide spatiotemporal attention with denoising. In this work, they consider, a simple yet effective, neuromorphic model called Leaky Integrate-and-Fire that makes use of the temporal information that decays over time. They incorporated a learnable time constant to be used in LIF and a thresholding-based efficient denoising technique. The experiments show a significant improvement over their predecessor, Spikformer.

**Strengths:**

Originality
- Incorporating spiking neural networks with the vision transformers is a very recent yet challenging issue. The authors are expanding on the work of Spikformer [1]. Even that work is very recent. The main objective of the paper is to make efficient spiking transformers that take advantage of temporal information.  Previous work is lacking in this regard; however, the authors have tackled this issue and improved the approach. They also involve a denoising technique for the accumulated spatiotemporal maps since they do not use a rather complicated function Softmax. Therefore, I believe this work is reasonably new and the concept itself deserves more attention.

Quality
- Spiking neural networks, more or less, mimic biological networks. Of course, it all depends on the biological plausibility of the models. For example, leaky integrate and fire models are very efficient neuromorphic models; however, compared to the Hodgkin-Huxley model, they are comparatively less biologically plausible. Thus said, current neuromorphic hardware does take advantage of the concept of LIF models to develop an SNN-based chip, and these models are not just space-efficient but also energy-efficient.
- Given the progression in transformers, the authors of this paper have shown that making transformers more biologically plausible can indeed have significant advantages over the other neuromorphic approaches. They do so by comparing against the most recent Spikformer [1], and other neuromorphic approaches with variable time steps.
- Specifically, the authors have integrated learnable time-constant (tau), which is a very important parameter in LIF that decides the significance of the input at the current time-step to the future ones. Further, to make the spatiotemporal attention mechanism less computationally expensive, the spiking networks have discarded Softmax masking, instead, the authors have used a threshold-based masking technique that significantly reduces the computational overhead.
- The authors have shown significant improvement over the other baseline approaches, and have done extensive ablation studies.

Clarity
- The paper is well-written and straightforward. The objective is clearly stated and tackled reasonably well throughout the paper.

Significance
- Since this approach is tackling a very recent and significant challenge of making neural networks more energy-efficient using spiking neural networks, it could have a good impact on the current research.

References:
1. Zhou, Zhaokun, Yuesheng Zhu, Chao He, Yaowei Wang, Shuicheng Yan, Yonghong Tian, and Li Yuan. "Spikformer: When spiking neural network meets transformer." arXiv preprint arXiv:2209.15425 (2022).

**Weaknesses:**

Concerns:
- There are a few things that the authors discussed in the paper; however, are not fully tackled.
  - Spiking networks by nature use less energy to function since they take advantage of the temporal coding. The authors have discussed this and it is one of the core reasons the community is interested in this, it would be interesting to perform a comparative analysis on hardware to test this approach. The authors have only discussed the accuracy of the approach.
  - Another thing would be space complexity. There should also be a comparative analysis of the space complexity of the approach.
- To get a better idea of the significance of the approach, comparative analysis on large benchmarks such as ImageNet would also be interesting to observe.
  - There are many hyperparameters involved: 𝜭, u, t, etc.

**Questions:**

The approach signifies the importance of spiking transformers; however, to fully exploit the potential of this approach, I believe, there are certain things to be considered:
- As mentioned in the weakness section, the paper does lack a rather important comparative analysis of energy-efficiency as mentioned. Please consider that as a measure of performance since it is one of the most crucial properties of the SNNs.
- I believe, with further analysis of the efficiency, this paper could be a good contribution to the vision community.

---

> ### Author Response · Authors · 2023-11-23
> **Thank you for your insightful review!**
>
> Thanks for your thoughtful and insightful questions!
>
> Q1: Spiking networks by nature use less energy to function since they take advantage of the temporal coding. The authors have discussed this and it is one of the core reasons the community is interested in this, it would be interesting to perform a comparative analysis on hardware to test this approach. The authors have only discussed the accuracy of the approach.
>
> A1: We provided a detailed hardware energy analysis in the general responses at the very beginning.
>
> Q2: Another thing would be space complexity. There should also be a comparative analysis of the space complexity of the approach.
>
> A2: We provided a detailed space complexity analysis in the general responses.
>
> Q3: To get a better idea of the significance of the approach, comparative analysis on large benchmarks such as ImageNet would also be interesting to observe.
>
> A3: We conducted an experiment on ImageNet-100 and presented the results in the general responses.

---

### Author Response · Authors · 2023-11-23
**General responses to reviewers (part1)**

We'd like to thank all reviewers for providing detailed and constructive feedback. We first provide general responses to several major issues raised during the review, and then answer individual questions from the reviewers.

A. Number of parameters

The vanilla Transformer model is primarily composed of Multi-Head Self-Attention mechanisms, Feed-Forward Neural Networks (FFNNs), and a series of Normalization Layers.
1. Multi-Head Self-Attention Mechanism: The parameters in each head are determined by weight matrices used to linearly transform the input into queries ($W_Q$), keys ($W_K$), and values ($W_V$) and output matrix ($W_O$). If the input dimension is \(d_{model}\) and the dimension per head is \(d_{qkv}\), then the number of the parameters of Multi-Head Self-Attention with $n_{heads}$
heads $N_{SA}$ is
\begin{equation}
    N_{SA} = N_{W_O} + N_{W_{QKV}} = (d_{model}+1)d_{model} + 3n_{heads}(d_{model}+1)d_{qkv}
\end{equation}
Since $d_{qkv} = d_{model}/n_{heads}$, the final number is
\begin{equation}
    N_{SA} = 4(d_{model}+1)d_{model}
\end{equation}
2. Feed-Forward Neural Networks: FFNNs consist of two linear transformation layers, typically interspersed with a ReLU activation function. The first layer expands the dimension from \(d_{model}\) to \(d_{ff}\), and the second layer reduces it back to \(d_{model}\). Therefore, the total parameter count for FFNNs is
\begin{equation}
    N_{FF} = (d_{model}+1)d_{ff} + (d_{ff}+1)d_{model}
\end{equation}
3. Normalization Layers: These layers generally have a smaller number of parameters, as they primarily include scale and bias parameters, typically is
\begin{equation}
    N_{Norm} = d_{model}*2
\end{equation}
In summary, a standard transformer with $L$ blocks contains
\begin{equation}
    N_{Trans} \approx L(4d^2_{model} + 2d_{model}d_{ff})
\end{equation}

In our DISTA, the parameters include:

1. Neuron-level spatiotemporal attention:  N denotes the number of tokens
\begin{equation}
    N_{neuron} = LN(6d_{model} + d_{ff})
\end{equation}

2. Network-level spatio-temporal attention: denotes T as the number of timesteps.

If there is no weight decaying:
\begin{equation}
    N_{network}= LTd^2_{model}
\end{equation}
If there is weight decaying: (described in "Parameter reduction for spatiotemporal attentions")
\begin{equation}
    N_{network}= Ld^2_{model}
\end{equation}

3. Spatio-temporal attention denoising:
\begin{equation}
    N_{denoising} = L
\end{equation}

B. Parameter reduction for spatiotemporal attention

The presented network-level spatiotemporal attention mechanism requires weight parameters that realize attention in both space and time. To manage the use of weights and the corresponding space complexity, we propose two distinct policies with two different space complexities. Under the first policy, we assign an independent set of trainable weights for each timestep, resulting in a total parameter count of  $O(LTD^2)$, where D indicates the feature number.  Under the second policy, we introduce a weight decay mechanism to reduce the space complexity of the weights. In this case, we use a set of weights for time 0, denoted by $W_0$. We decay $W_0$ over time based on a decay constant $\tau$ to define the weights for the subsequent time steps without introducing additional parameters: $W_t = e^{-t\tau}W_0$.  Notably, this approach maintains the parameter count at $O(LD^2)$, while still achieving comparable performance as demonstrated in our experiments.

---

### Author Response · Authors · 2023-11-23
**General responses to reviewers (part2)**

C. Energy Efficiency

We demonstrate the energy efficiency of the proposed DISTA approach via systematic evaluation.  We focus on the key computation overhead associated with attention maps by analyzing the total energy consumed by computing the attention outputs for a batch of images, where the batch size is 128.  We model the total energy dissipated by counting the number of accumulate operations used to realize Sparse-Dense matrix multiplications: $Energy$ = (1-$sparsity_{attn}$) $\times$ #$OPs_{AC}$ $\times$ $E_{AC}$. We characterize the 8-bit accumulation operation's energy consumption  $E_{AC}$ to be 0.019 pJ based on a commercial 28nm technology library. The energy analysis result is shown in the below Table. For the vanilla Spikformer, there exists an average attention map sparsity of 77.35\% for a 4 encoder-block transformer. Denoising the first two or three encoder blocks in DISTA increases the on-average sparsity to 86.42\%(+9.08\%) and 93.1\%(+15.75\%)  respectively with an improved accuracy of **+0.95\%** and **+0.91\%**. Thus, DISTA has an average energy consumption saving of **40.04\%** and **69.49\%** over Spikformer.


| Model               | Block Index | Sparsity (%)               | Energy Consumption (nJ) | Accuracy               |
|---------------------|-------------|-----------------------------|-------------------------|------------------------|
| Spikformer          | 0           | 78.3%                       | 6.48                    | 95.34%                 |
|                     | 1           | 74.2%                       | 7.71                    |                        |
|                     | 2           | 80.4%                       | 5.86                    |                        |
|                     | 3           | 76.5%                       | 7.02                    |                        |
| **DISTA(b=2)**      | 0           | 98.2%(+19.9\%)     | 0.54(-91.7\%) | 96.29%(+0.95\%) |
|                     | 1           | 96.4%(+22.2\%)     | 1.08(-86.0\%)  |                        |
|                     | 2           | 75.1%(-5.3\%)      | 7.44(+27.0\%)  |                        |
|                     | 3           | 76.0%(-0.5\%)      | 7.17\(+2.12\%)  |                        |
| **DISTA(b=3)**      | 0           | 97.6%(+19.3\%)     | 0.72(-88.9\%)  | 96.25%(+0.91\%) |
|                     | 1           | 97.3%(+23.1\%)     | 0.81(-89.5\%)  |                        |
|                     | 2           | 98.3%(+17.9\%)    | 0.51(-91.3\%)  |                        |
|                     | 3           | 79.2%\(+2.7\%)      | 6.22(-11.4\%)  |                        |


D. Results on ImageNet-100

We conduct an experiment on the ImageNet-100 dataset to benchmark our model's performance against prior works. For this experiment, we configure the model with a feature count of 128 and a patch size of $16 \times 16$. The training is carried out over 100 epochs. The initial learning rate is set at $5 \times 10^{-4}$, and the minimum learning rate is $5 \times 10^{-6}$, managed with a cosine learning rate scheduler. To ensure a fair comparison, we match the parameter count of our DISTA model with that of the conventional spikformer model, with both models having 1.86 million parameters. As shown in the Table below, the prior spikformer achieves a 66.69\% accuracy, and DISTA achieves **66.79\%(+0.4\%)** when enabling attention denoising(ADN) and achieves **67.92\%(+1.53\%)** when enabling both of ADN and weight decaying(our network-level spatiotemporal attention).

| Model                             | # Timesteps | # Tokens | # Features | Accuracy                 |
|-----------------------------------|-------------|----------|------------|--------------------------|
| Spikformer                        | 4           | 196      | 128        | 66.39%                   |
| DISTA (only enable ADN)           | 4           | 196      | 128        | 66.79% (+0.4%)  |
| DISTA (enable ADN + weight decaying) | 4         | 196      | 128        | 67.92% (+1.53%) |

---

### Meta-Review · Area_Chair_nvdC · 2023-12-05

**Metareview:**

This paper presents a novel spiking transformer model. There was general agreement between the reviewers that the work appears relatively incremental compared to published work. The authors tried their best to address the reviewers' comments during the rebuttal. Still, the work remains largely incremental compared to already published spiking transformer models, given that the reported improvements appear quite marginal. While the AC understands that there are differences between the proposed architecture and the work by Wang et al. (Spatial-Temporal Self-Attention for Asynchronous Spiking Neural Networks) brought up by one of the reviewers, the significance of the present work remains quite limited. Overall, the AC recommends this paper be rejected.

**Justification For Why Not Higher Score:**

The significance of the work appears too limited.

**Justification For Why Not Lower Score:**

NA

---

### Decision · Program_Chairs · 2024-01-16

Reject